# Unmet vaccination need among children under the age of five attending the paediatric emergency department: a cross-sectional study in a large UK district general hospital

Rachel Isba [iD],[1,2,3] Louise Brennan [iD],[1,2] Fiona Egboko,[1] Rhiannon Edge [iD],[1] Nigel Davies [iD],[1] Joanne Knight [iD][1]

¹Lancaster University, Lancaster, UK
²Emergency Department, North Manchester General Hospital, Manchester University NHS Foundation Trust, Manchester, UK
³Medical Services Division, Alder Hey Children's NHS Foundation Trust, Liverpool, UK

**Correspondence to**
Professor Rachel Isba;
rachel.isba@lancaster.ac.uk

## ABSTRACT

**Objective** To estimate vaccination coverage among children under the age of five attending the paediatric emergency department (PED) using tetanus and MMR vaccination as a proxy.

**Design** A cross-sectional observational study with a single data collection point for each participant.

**Setting** A single large PED in Greater Manchester, England.

**Participants** Children (under 5 years old) attending the PED during October 2021. Participation was 'opt-out' and parents/carers were given until the end of the following month to request that their child's data be excluded.

**Primary and secondary outcome measures** The primary outcome of interest was the percentage of children who were up-to-date with their routine childhood vaccinations at their time of attendance to the PED. Secondary outcome measures were the percentage of children who had received age-appropriate tetanus and MMR vaccination, and how these compared with local population data at the ages of 1, 2 and 5 years of age.

**Results** One-third of under-5s in this study had unmet vaccination need and were missing at least one dose of either MMR or tetanus-containing vaccine. In older age groups, many were missing their tetanus boosters and only 1 in 5 of those eligible had received two doses of MMR. Those in younger age groups had vaccination coverage levels comparable to the local data, but still below the target of 95%.

**Conclusions** Those children eligible for preschool boosters (tetanus and MMR2) appear to have considerable unmet vaccination need. While the pandemic has had an impact, the observation that MMR2 uptake is considerably lower than tetanus booster (when they are scheduled together) warrants further investigation. Catch-up campaigns for MMR2 should focus on this cohort of children and the PED may offer an opportunity for an intervention.

**Trial registration number** NCT04485624.

## INTRODUCTION

Vaccines are one of the great global health successes. Since their discovery more than

---

### STRENGTHS AND LIMITATIONS OF THIS STUDY

⇒ This study is timely given the pandemic's impact on routine childhood vaccination.
⇒ A population has been identified with considerable unmet vaccination need who might benefit from an intervention not currently offered.
⇒ A limitation is that the work was undertaken at a single centre and there were constraints with data collection and quality.
⇒ A proxy for overall vaccination status was used, leading to a possible underestimation of unmet vaccination need.

---

300 years ago, vaccines have saved countless millions of lives,[1] reduced the incidence of dozens of diseases and even lead to the eradication of smallpox.[2] In countries with access to them, vaccines have also played a key part in the control of the SARS-CoV-2/COVID-19 pandemic.[3]

However, prepandemic, in the UK, uptake of routine childhood vaccination had fluctuated[4] and coverage lagged behind some of our European peers for common vaccine-preventable diseases such as measles.[5]

Every year in England, millions of children and young people (CYP) attend hospital (secondary or tertiary medical care).[6] Attendance is often with relatively minor illnesses and injuries, many of which could be better managed elsewhere. However, despite numerous initiatives to redirect these CYP, hospital attendances (prepandemic) had increased year on year.[6] The pronounced decrease in paediatric emergency department (PED) attendances seen early in the pandemic[7 8] was reversed in 2021, as lockdowns and other restrictions were eased, with attendance exceeding prepandemic levels

and a change in some of the seasonal patterns of illnesses presenting to the PED.[9]

In addition to their primary reason for presentation, CYP attending the hospital may have lower than average levels of health and well-being, additional unmet health need (eg, dental health), or not be able to engage with preventive elements of routine healthcare (eg, vaccination) for a myriad of reasons. A hospital attendance or admission might therefore offer an opportunity to intervene. A recent pilot study showed that time during a PED department attendance could be used to deliver a public health intervention and that this was both feasible (within the constraints of the department) and acceptable (to all stakeholders including CYP, parents and carers, and staff in the PED).[10]

If any child or young person who had not received their age-appropriate routine vaccinations could be identified during a PED attendance, clinicians might (should it be clinically/situationally appropriate) be able to offer one or more tailored interventions to address this unmet vaccination need. The benefits of such an approach are numerous and include:

► Decreasing mortality and morbidity from vaccine-preventable diseases, by ensuring
  – Individual and population coverage for diseases that cannot spread person–person, for example, tetanus.
  – Higher levels of population coverage for non-epidemic diseases that can be spread person–person, for example, hepatitis B.
  – Herd immunity for diseases that can easily spread person–person and can cause outbreaks, for example, measles.
► A decrease in un-needed treatment in the case of individual exposure in the absence of an accurate vaccination history at the point of treatment, for example, a tetanus-prone wound in the PED.
► A reactive response to outbreaks, for example, mumps; epidemics, for example influenza; and pandemics, for example, SARS-CoV-2.
► Improving coverage of targeted vaccination programmes, for example, seasonal influenza.

In the UK the routine immunisation schedule is commissioned nationally. Vaccinations are free and parents/carers of children are invited and reminded about vaccinations. Preschool vaccinations are done through general practice, while school-age services are delivered through School Immunisation Teams. Immunisations are recorded in Primary Care Records (maintained by general practitioners) and the Child Health Information Service. In some regions these vaccination records are not accessible in secondary care, however, some areas have access to immunisation information via summary care records (SCRs) which are centralised NHS computer systems containing information such as patient medications and allergies.[11] Regardless of whether SCRs are available, the vaccination status of a child should be checked with the caregiver as part of routine care within

these settings.[12] While vaccines are not mandatory, they are recommended, and the Department of Health and Social Care take responsibility for the vaccination strategy and initiation of promotion/uptake campaigns.[13] In its 2021 Health Equity Audit of the National Immunisation Programme,[14] Public Health England (now the UK Health Security Agency) stated 'Equality in immunisation is an important way to address health inequalities' and reported that while the Immunisation Programme had achieved high coverage in the population as a whole, within subpopulations there still existed 'avoidable inequalities'. While the reasons for these inequalities are complex, 'institutional' and 'policy' factors play a role.[14] Addressing these factors by offering vaccination at the point of PED attendance may preferentially improve vaccination uptake among those experiencing avoidable inequalities.

Previous work to improve vaccination uptake via interventions delivered in secondary or tertiary care (see[15] for overview) has shown that vaccination coverage in CYP attending hospital settings is generally lower than in the general population. However, the vaccinations under investigation, location of presentation within the hospital, and way in which vaccination data were verified, varies considerably. In the previous study,[15] for all due/overdue vaccinations (not a measure currently available in England), baseline coverage ranged from 44%[16 17] to 89%,[18] with little difference by setting and a trend for lower coverage in older studies. For influenza, baseline coverage was lower, ranging from 25%[19] to 50.5%.[20]

The overall aim of this work was to answer the question: do children (aged <5 years) attending the PED have lower levels of vaccination coverage than their peers in the general population? This allowed us to assess the unmet vaccination need of children attending the PED.

## METHODS

This was a cross-sectional observational study with a single data collection point for each child. The study was registered as an observational/non-interventional study on the clinicaltrials.gov website.

### Consent

Prominently displayed posters relating to the study were put up around the PED during the month of data collection, with flyers handed out in triage, and participant information sheets with opt-out consent forms available on request. Nursing staff in the PED were briefed so that they could support queries from parents/carers. Parents/carers could choose for their child to opt-out by filling in the form or contacting a member of the research team within a month of attendance.

### Setting

Data were collected from a single, dedicated colocated PED, in Greater Manchester, in the North West of England.

## Participants

Participants were children (under 5 years old) attending the PED during October 2021. CYP in Manchester have lower than average levels of health and well-being, around a third live in low-income households, and 1 in 100 of them reside in care.[21]

## Patient and public involvement

Reading materials were designed to be read by the parent or caregiver. Following patient involvement in a previous study[10] all reading materials were designed for a reading age of 7–9 years old, in order to be as accessible as possible.

## Vaccination schedule

The routine NHS vaccination schedule[22] recommends tetanus-containing vaccines (given as multi-component vaccines) are given at 8, 12 and 16 weeks ('primary course') and at around 3 years 4 months (as part of the so-called 'preschool booster'). MMR1 is due around 1 year of age and MMR2 is given at 3 years 4 months, alongside the preschool booster.

## Data collection

A list of dates of birth and hospital numbers was generated for all attendees during the month of October 2021. Duplicates (where the same child may have attended more than once in the month of interest) were removed. Exact age at presentation was not available, so was calculated based on age in completed months on 1 October 2021. Any vaccination before the end of October 2021 was included when checking status. This approach was taken to ensure that the data tended to over-estimate rather than under-estimate coverage, for example, at the lower age limits.

Data were extracted from individual SCRs by RI and anonymised before analysis. In Greater Manchester, SCRs contain information on the child's health and vaccination status. Extraction took between 2 and 5 min per record (depending on the quality of network access/connectivity). The quality and presentation of the data within the SCR was also very variable and this finding is presented elsewhere.[23]

## Sample size calculation

The MMR vaccine was chosen for the sample size calculation as this is the vaccine that has attracted the most controversy over the past decades (resulting in lower than target coverage).

At the time the sample size calculation was carried out, the most recent data for MMR2 coverage at 5 years were available for the year 2018/2019.[24] In England, coverage for MMR2 was 86.4% and for Manchester, coverage for 5 years old was 82.1%. The sample size calculation was carried out using STATA V.16[25] and a comparison made between population prevalence (between the 'PED' and 'general Manchester' populations). The sample size calculation was based on the difference between the population prevalence in Manchester of 0.82 and in the study PED of 0.77 (ie, assumed to be 5% less than the general population), with the required power at 0.8, using a two-sided test, with alpha level set at $p<0.05$. This resulted in an estimated sample size for a one-sample proportion test (Wald z test) of 577.

## Outcomes

### Primary

The primary outcome of interest was the percentage of children aged >2 months (to reflect the start of the immunisation schedule in the UK) and <5 years (the age at which UK Cover of Vaccination Evaluated Rapidly (COVER) data assesses vaccination uptake) who were 'up to date' with all their age-appropriate tetanus-containing and MMR vaccinations at the time of PED attendance (this was used as a proxy for overall vaccination status).

Children were coded as 'up to date' with their vaccinations if they had received all tetanus and MMR vaccinations for which they were eligible (based on their age) by 31 October 2021.

### Secondary

Secondary outcomes of interest were age-appropriate tetanus-containing vaccination coverage, and uptake of MMR1 and MMR2. Specifically:

► >2 months and <5 years who had received all their age-appropriate tetanus vaccinations at the time of PED attendance.
► >12 months and <5 years who had received at least one dose of MMR at the time of PED attendance.
► Between 2 and <5 years who had received MMR1 by their second birthday.
► Between 3 years 4 months and <5 years who had received two doses of MMR at the time of PED attendance.
► Between 3 years 4 months and <5 years who had received MMR2 by their fifth birthday.

### Statistical analysis

Descriptive statistics were prepared for primary and secondary outcomes. We compared our data to data published for the year April 2021–2022 in the COVER programme.[4] The COVER programme publishes quarterly and annual vaccination coverage statistics for children aged 12 months, 24 months and 5 years in the UK. Data from 2021 to 22 was chosen as they were the most recent published data and corresponded with the study period.

We mirrored the methodology used for the COVER data[26] using number of eligible persons immunised as the numerator and total number of persons in the eligible population as the denominator, to obtain a comparable sample from our study in the PED. We compared our data at 12 months, 24 months and 5 years with the publicly available data for the Lancashire and Greater Manchester footprint. This geography includes the majority of the catchment area for the PED in our study. Due to the age range in our sample (2 months–5 years) it was not possible to mirror the 5-year-old denominator methodology used

for national samples (total number of children reaching their fifth birthday within the evaluation dates), we therefore used children aged 4 years 0 months–4 years 12 months as a proxy for 5-year-old data.

We compared the coverage figures for children in our study with local and national data obtained from the COVER programme. $\chi^2$ tests were used to examine differences in proportions and generate significance levels.

## RESULTS

A total of 1450 children under the age of 5 years attended the PED in October 2021. Of these, 113 were under 2 months old (the age at which first tetanus-containing vaccines are given) and were therefore excluded. For the remaining 1337 children, records were available for 1223 (91%) of them and are included in subsequent analyses. Records were unavailable if it was not possible to see the vaccination data, for example, children living outside Greater Manchester had no GM SCR. No parents/carers chose to 'opt-out' of the study. Children in the under-5s age group made up around 60% of all attendees to the PED (aged <16 years at the time of presentation) in October 2021.

## Participants

As the main variable of interest was attendance at the PED, no data beyond age were collected. The age distribution of children across ages was roughly uniform (online supplemental file 1).

### Tetanus-containing and MMR combined as a proxy for overall vaccination status

At the time of their presentation to the PED, two-thirds of the 1223 children had received all of their age-appropriate tetanus-containing and MMR vaccines (n=807; 66.0%). This effectively equates to 416 missed opportunities to identify under-vaccinated children during their attendance within this sample.

Vaccination status varied by age band, with younger age groups tending to have higher levels of coverage (figure 1, online supplemental file 2). There were notable 'dips' within the age bands where vaccines were due, for example, MMR1 falls within the band 12 months to <15 months and coverage in this band was 70.3%, compared with 96.1% in the preceding band and 80.3% in the subsequent band. As a group, 4 years old had very low levels of tetanus and MMR coverage, with just 1 in 5 of them having received a full set of primary and booster doses of tetanus plus two MMRs. Twenty-nine children in the sample had no vaccinations recorded, for example,

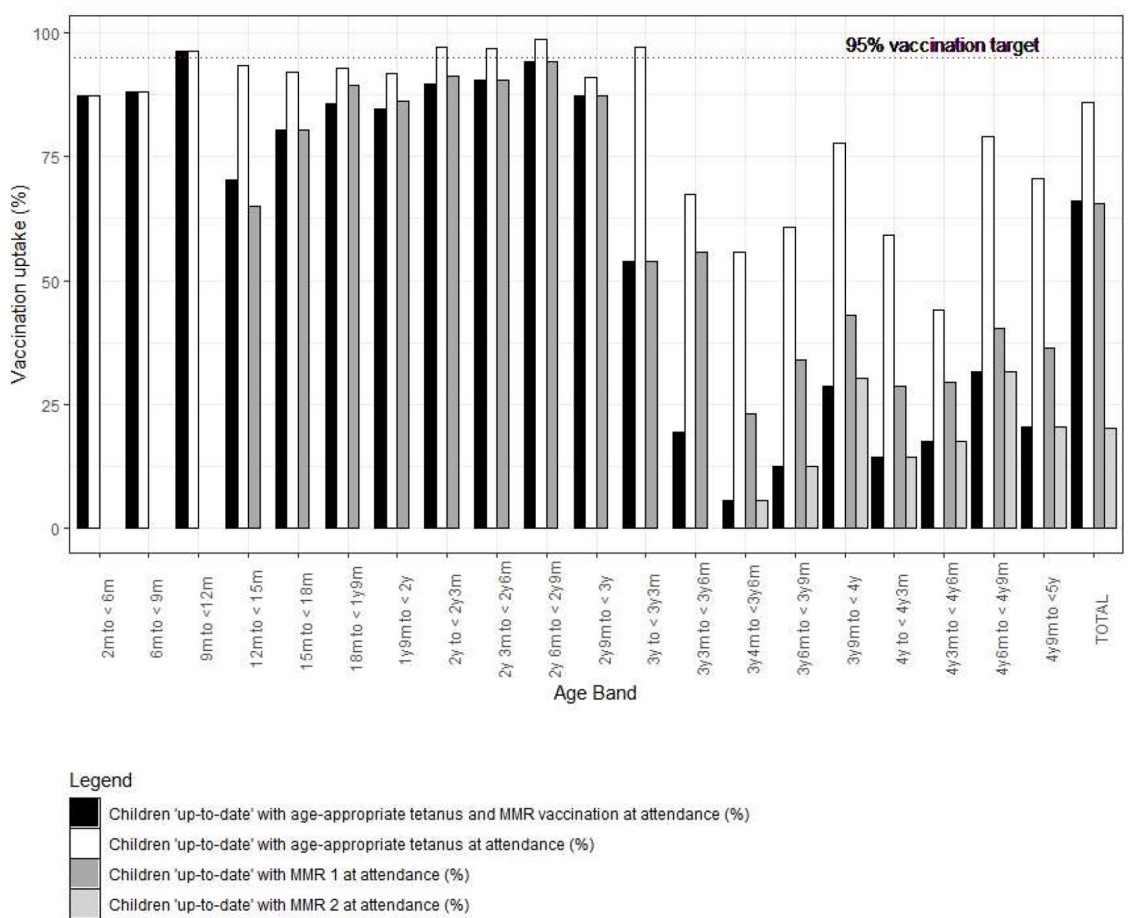

**Figure 1** Level of vaccination coverage by age. Data to support figure 1 is included in online supplemental file 2.

the parent/caregiver had not given consent for vaccinations and this was recorded in the SCR.

### Tetanus-containing vaccination

In the sample of children included in this study, 85.9% had received all age-appropriate tetanus-containing vaccinations at the time of presentation at the PED. Again, this varied by age group with higher levels of coverage present in children eligible only for their primary tetanus vaccines and coverage dropping in children from the age of 3 years 4 months when the 'preschool' booster vaccine is due.

### MMR1

During October 2021, 976 children attending PED were eligible for their first dose of the MMR vaccination. Of these, 640 (65.6%) had received the immunisation, meaning one-third of children had not received their MMR vaccine despite being eligible (figure 1).

### MMR2

Of the 339 eligible children (older than 3 years and 4 months), only 68 had received their second MMR vaccination (20.1%). This was lowest in the youngest age band (3 years 4 months–3 years 6 months), with only 5.6% already having received the vaccination before their PED attendance. However, this rate remained low, even in the oldest children, for example, only 14% of the children aged 4 years–4 years and 3 months had received their MMR2 vaccine, and 20.5% in children aged 4 years and 9 months and 5 (figure 1).

### Comparison with national COVER data

Comparisons between publicly available data for the catchment area (Greater Manchester and Lancashire) and the children in our sample found no statistically significant differences in coverage at 12 and 24 months (table 1) or for the receipt of the full primary course of tetanus-containing vaccines by the age 5 years.

However, uptake of MMR1, MMR2 and the tetanus-containing booster (all by the age of 5 years) were all significantly lower in our sample of children than the general local population (p<0.0001).

Finally, we compared the coverage in our sample to data for England at age 5 years. The children in our study had a higher proportion of coverage for the primary tetanus containing vaccine than the English sample (0.9% higher), however, this was not statistically significant. A significantly lower proportion of children in our study were vaccinated with MMR1, MMR2 and the tetanus-containing booster vaccine than in the 2021–2022 English cohort (60.4%, 65.3% and 19.9% less, respectively; all p<0.0001). While there may be lag noted in some of our data (where children have a low rate of coverage in the age bands immediately after the age at which their vaccine is first due), this is unlikely to impact comparisons with the COVER data as the comparator points are not around the age at which vaccines are due.

### DISCUSSION

The WHO recommends that at least 95% of children should be vaccinated against diseases preventable by immunisation to ensure elimination and control.[27] Our data suggests that the children attending this PED in Greater Manchester were below this recommendation for MMR1 and MMR2, and for the tetanus-containing booster.

When comparing our PED population to their peers in the general population we found that there were no statistically significant differences in coverage at age 12 and 24 months. However, uptake of MMR1 and MMR2 and the tetanus containing booster at age 5 were significantly lower in the PED than in the general population. This study supports previous findings[14] and suggests that the PED is an opportune location for vaccine interventions.[10]

### Estimate of 'missing' tetanus and MMR vaccinations in a year

To estimate the total number of 'missing' vaccinations among children under the age of five attending the study PED in a year, we assumed that the attendance figures for

**Table 1** Proportion (and number) of children vaccinated and comparison of sample data (PED) with 2021–2022 COVER data for Lancashire and Greater Manchester (L&GM)

| | 12 months | | 24 months | | 5 years | |
|---|---|---|---|---|---|---|
| | L&GM | PED | L&GM | PED | L&GM | PED* |
| Tetanus-containing primary course vaccine† | 91.5 (44 195) | 92.4 (283) | 93.7 (47 600) | 96.0 (243) | 95.1 (53 541) | 96.1 (177) |
| MMR1 | | | 90.3 (43 615) | 90.9 (230) | 94.6 (53 260) | 34.2‡ (63) |
| MMR2 | | | | | 87.0 (48 981) | 21.7‡ (40) |
| Tetanus-containing booster§ | | | | | 84.6 (47 630) | 64.7‡ (119) |

The cells are just shaded to indicate that there wouldn't be data for those cells as the vaccines aren't given yet.
*Children aged 4 years 0 months–4 years 12 months used as a proxy for 5 years old.
†COVER data reports DTaP/IPV/Hib/HepB3.
‡Statistically significant difference in proportions of children vaccinated between L&GM and PED p<0.0001.
§COVER data reports diphtheria, tetanus, polio, pertussis, Hib booster.
COVER, Cover of Vaccination Evaluated Rapidly; PED, paediatric emergency department.

October 2021 were representative of a typical month in 2021 (recognising that pandemic-related restrictions and impacts were ongoing throughout 2021 and PED attendance would usually fluctuate across the year with higher rates in winter and lower rates occurring in summer). Scaling up, we estimated that a total of 4992 children would have attended the study PED in 2021 with at least one missing tetanus or MMR vaccination. Specifically, if PED teams were equipped to administer catch up vaccinations (and all children were eligible and all parents/carers consented), up to 4032 MMR1s, 3252 MMR2s and 1404 tetanus boosters could potentially have been administered during 2021. Vaccines could also have been given to start, continue or complete 960 primary tetanus courses (which would have required varying numbers of additional doses depending on the age of the child). It is recognised that these are approximations and could be overestimating, for example, as some children attending the PED will be too unwell to have an intervention.

### Principal findings

In the month of October 2021, 416 children (34% of those attending) were not up-to-date with their routine vaccinations at the time of their PED attendance. This could equate to up to 5000 potential vaccination moments in a single large PED—a currently unused opportunity to address unmet vaccination need, particularly among those experiencing vaccination inequity. MMR vaccination uptake in this PED-attending population was particularly low in older age groups, and significantly lower than the general population. While some of this is likely due to pandemic-related disruption (eg, lockdowns), the finding that tetanus booster uptake, while low, was not as low as MMR2 uptake (despite them being offered during the same appointment), suggests that there may be something else contributing to the very low MMR2 numbers, and this would warrant further investigation in future studies.

### Strengths and weaknesses of the study

While not in the original study design, a particular strength of this work is that it captured data within the pandemic and has provided additional insights into subpopulations of children who are now very under-immunised. Due to the pandemic, the study was restricted to a single site as the second study site (a local children's hospital) was not able to open to recruitment.

As with all studies of this type, there were issues around data quality and access, with around 10% of potential participants excluded due to inaccessible data. There were some obvious errors and inaccuracies within the SCR data (eg, too many vaccines recorded or 'immunisations given' with no further detail), which may have contributed to over-estimation or under-estimation of vaccination status.

This study took place in an unusual data ecosystem where access to a SCR (and therefore vaccination status) was possible. This study may not be reproducible elsewhere

and recommendations for interventions to investigate vaccination status within the PED would require access to this information. Access to reliable vaccination data in acute care settings remains an issue.

A proxy was used for overall vaccination status (tetanus and MMR combined) which is not ideal, but will only have led to an underestimate of the unmet vaccination need of children presenting to the PED. During the course of the work it became apparent that trying to find meaningful comparators for the study data would be tricky as national data are presented by individual vaccine type with no data available as to children's overall vaccination status, that is, whether or not they are up-to-date with all their age-appropriate vaccinations. For the comparison at 5 years of age, we used a slightly younger cohort than the COVER data. However, as vaccination coverage rates locally, regionally, and nationally usually only increase by around 5% between the ages of 2 and 5 years,[4] this is not likely to have led to a change in the statistical significance of the results, as the coverage at age 5 for everything except primary tetanus was so low.

### Meaning of the study and implications

Under-5s attending the PED have unmet vaccination need, particularly among those eligible for MMR2 and tetanus boosters, and uptake of these so-called 'preschool boosters' differed by vaccine type.

Although there was no difference in vaccination coverage between the younger PED attendees and their peers in the local general population, they still had uptake levels below the 'ideal' of 100% and mostly below the 95% target suggested by the WHO.

If one or more interventions were available to deliver to these under-vaccinated children during a PED attendance, hundreds of thousands of additional potential vaccine opportunities would be available to practitioners every year. Any future local or national campaigns, for example, 'catch-up' for MMR, could look to use the PED as a potential site for delivery of vaccinations.

### Future research

Future research will look at those aged 5 and older, to see if the patterns of unmet vaccination need are sustained in these other age groups, or if they are a pandemic-related phenomenon (or a combination of both). The cohort identified in this study as being under-immunised for tetanus and MMR might benefit from a 'catch-up' campaign, and the PED would seem to be a potential additional location for this to be delivered. Future work will also look to codesign one or more interventions to support MMR catch-up in this age group, as well as exploring the potential for the development of other vaccination-focused programmes that could be delivered in the case of an outbreak, for example, mumps, or for routine vaccination for example, influenza.

Having identified this under-immunised population attending the PED, it may be that there are other under-immunised CYP attending other settings within the

hospital who might benefit from the offer of a vaccination-focussed intervention. Identification of these populations, for example, those attending outpatient clinics, would facilitate more opportunities to offer vaccination interventions as part of routine hospital care.

**Contributors** RI conceived of the study, led on the ethics application process, undertook the fieldwork, analysed the data, was involved in all stages of the preparation of this manuscript and is guarantor author. LB analysed the data and was involved in all stages of the preparation of this manuscript. FE supported data extraction and was involved in later drafts of the manuscript. RE supported data analysis and was involved in later drafts of the manuscript. ND and JK were involved in oversight of the project and were involved in later drafts of the manuscript.

**Funding** RI was supported by a Health Education England Topol Digital Health Fellowship from February 2021 for a year.

**Competing interests** None declared.

**Patient and public involvement** Patients and/or the public were involved in the design, or conduct, or reporting, or dissemination plans of this research. Refer to the Methods section for further details.

**Patient consent for publication** Not applicable.

**Ethics approval** This study involves human participants and full prospective ethics approval was obtained from the North West—Greater Manchester East Ethics Committee (IRAS reference 278815, REC reference 20/NW/0423). Following an earlier part of the work, an ethics amendment was submitted and approved to move to 'opt-out' consent for this part of the project (Amendment 1, substantial, non-CTIMP, approved 26 August 2021). Participants gave informed consent to participate in the study before taking part.

**Provenance and peer review** Not commissioned; externally peer reviewed.

**Data availability statement** Data are available upon reasonable request. Relevant data may be requested from the lead author and will be provided in anonymised format as soon as legally and ethically possible.

**ORCID iDs**
Rachel Isba http://orcid.org/0000-0002-2896-4309
Louise Brennan http://orcid.org/0000-0003-0377-6720
Rhiannon Edge http://orcid.org/0000-0001-5349-783X
Nigel Davies http://orcid.org/0000-0002-4136-5300
Joanne Knight http://orcid.org/0000-0002-7148-1660

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
