## [Reviewer comments · BMJ Open]

ARTICLE DETAILS

TITLE (PROVISIONAL)	Unmet vaccination need amongst children under the age of five attending the Paediatric Emergency Department: a cross-sectional study in a large UK district general hospital
AUTHORS	Isba, Rachel; Brennan, Louise; Egboko, Fiona; Edge, Rhiannon; Davies, Nigel; Knight, Joanne

VERSION 1 – REVIEW

REVIEWER	Vaillant, Vera University Hospitals Giessen and Marburg Campus Giessen
REVIEW RETURNED	19-Feb-2023

GENERAL COMMENTS	Isba et al. present a cross-sectional study on the vaccination rates of children under the age of five attending the paediatric emergency department (PED) in a large UK district general hospital. The article is timely as we need to identify under-immunized patients and find suitable settings to create opportunities for catch-up interventions. Methods: Why did you choose the age-range of 2 months to 5 years if it is not possible to mirror the 5-year-old denominator methodology used for national samples you wanted to compare your data to? Particularly in the absence of exact age at PED consultation the rationale for the fairly large quantity of different age groups should be explained. Page 12, line 50: "Data from 2021-22 was chosen as they were the most recent published data." The study period also corresponds to this data. Results: Page 13, line 33: "Of these, 113 were under 2 months old (...)so were excluded." Please check this sentence. How many parents used the opt-out option? What were the other reasons for missing data? Page 14, line 27: "Twenty-nine children in the sample appeared to be wholly unvaccinated." Was there no data on any vaccinations in the summary care records or is this an assumption due to missing tetanus and MMR vaccinations? Page 14, line 44: At the end of the paragraph there is an unfinished sentence beginning with "MMR vaccination"
---

	Page 15, line 14: Children aged 4 years to 4 years and 3 month are not the oldest children in the cohort. Perhaps you could say that in the children older than 4 years 14-32% had received their MMR2 vaccine. The estimation of “missing” tetanus and MMR vaccinations in a year seems unreliable when PED attendance usually fluctuates across the year and there were additional pandemic-related anomalies. Moreover, multiple visits of one patient or contraindication for vaccination such as severe underlying disease or acute clinical condition were not considered. Page 15, line 48: You repeat the statement from line 9-14. Discussion Page 17 line 23-30: I appreciate, that you want to illustrate the currently unutilized opportunity to address unmet vaccination needs in PEDs. But the uncertainty in your estimation of 5,000 potential vaccination moments in one year (period not mentioned) is not mentioned (see above). Page 17, line 30-42: You explained, that vaccination rates are lowest in the age groups immediately after the recommended vaccination age and that there seems to be a catch-up effect with time. This may need to be considered for the comparison of your data and the COVER data. Page 18, line 5: I think that it may be interesting to discuss that the available methods to screen immunization status will vary depending on the country. Recall of immunization status is unreliable and handheld immunization records will most likely be unavailable in many PED consultations. In countries with Immunization registries or summary care records provide the opportunity to screen immunization status in a way that is not available in my country. I therefore need more information to understand what comprises inaccessible data? Were summary care records inaccessible?
--	--

REVIEWER	Brandstetter, Susanne University Children's Hospital Regensburg
REVIEW RETURNED	07-Mar-2023

GENERAL COMMENTS	Thank you for the opportunity to review this interesting manuscript. The authors address an important topic for child health. The manuscript is clearly structured and well written. However, I believe it could profit from some additional work. Please find below my suggestions: Introduction: Since BMJopen addresses also international readers who might not be familiar with the (child) health care system in the UK, it might be helpful to provide more detail on the organization of vaccination campaigns (Are vaccinations mandatory? Are parents invited? Are there incentives? Do parents have to pay? etc.) and the documentation of vaccinations (Who does the documentation? Who keeps the health records? Who can access them? etc.). Is checking of vaccination status routinely part of treatment in PED?
---

	The benefits of using the PED for vaccinations enumerated in the introduction go very far. I suggest to focus more on the immediate benefits and on topics which are addressed by this study. Page 8: “Within the scoping review...”. Which scoping review? The research question aims at demonstrating lower vaccination coverage in the PED compared with the general population. Please explain why you were interested in the difference of vaccination coverage. Methods: Please provide some information on the structure and data quality of health records. Sample size calculation: What is the source for the PED prevalence of 0.77? Outcomes are described twice in the methods section. Please consider removing some of the information. Results: Participants: “was roughly uniform.” Please provide data to support this statement. Estimate of “missing” tetanus and MMR vaccinations in a year: Please consider moving this paragraph to the discussion section. Table 1: Please provide Ns. Figure 1: bars for 3y3m to 3y6m: is the data correct? For most other age groups the black bars (children with tetanus and MMR vaccination) roughly correspond to the lowest bar within this age group (i.e. MMR vaccination). Figure 1: age band “total”: are all categories (bars) applicable for the total sample? Discussion: The discussion covers many interesting and important topics. However, I miss the discussion of your main research question (difference in vaccination coverage between PED and general population). The study’s findings on great differences in tetanus and MMR vaccination prevalences is interesting. Do you have any ideas why MMR vaccination rate are lower? The study period included the COVID-19 pandemic, including reduced access to health care in lockdown periods. Can you see potential lockdown effects in some age strata of your data?
--	---

VERSION 1 – AUTHOR RESPONSE

RESPONSE TO REVIEWERS	
Comment/ amendment	Response
Why did you choose the age-range of 2 months to 5 years if it is not possible to mirror the 5-year-old denominator methodology used for national samples you wanted to compare your data to? Particularly in the absence of exact age at PED consultation the rationale for the fairly large quantity of different age groups should be explained.	Thank you for your comment. Additional information has been added the outcomes section to clarify this point.

Page 12, line 50: "Data from 2021-22 was chosen as they were the most recent published data." The study period also corresponds to this data.	Thank you for your comment. Your point has been added to the manuscript.
Page 13, line 33: "Of these, 113 were under 2 months old (...)so were excluded." Please check this sentence.	Thank you for your comment. This sentence has been amended to make clearer.
How many parents used the opt-out option? What were the other reasons for missing data?	Thank you for your comment. No parents chose to opt out of the study. A sentence has been added to the results section to clarify this. Data was deemed missing if it was not possible to see the vaccination record, e.g. if children were from outside Greater Manchester then no summary care record would be available. This has also been clarified in the results section.
Page 14, line 27: "Twenty-nine children in the sample appeared to be wholly unvaccinated." Was there no data on any vaccinations in the summary care records or is this an assumption due to missing tetanus and MMR vaccinations?	Thank you for your comment. This has now been clarified.
Page 14, line 44: At the end of the paragraph there is an unfinished sentence beginning with "MMR vaccination"	Thank you for your comment. This has been removed.
Page 15, line 14: Children aged 4 years to 4 years and 3 month are not the oldest children in the cohort. Perhaps you could say that in the children older than 4 years 14-32% had received their MMR2 vaccine.	Thank you for your comment. We have added in the rate for the oldest cohort of children in response.
The estimation of "missing" tetanus and MMR vaccinations in a year seems unreliable when PED attendance usually fluctuates across the year and there were additional pandemic-related anomalies. Moreover, multiple visits of one patient or contraindication for vaccination such as severe underlying disease or acute clinical condition were not considered.	Thank you for your comment. We have added further detail to reiterate that these are approximations that may be overestimates. In response to a later comment we have also moved this section to the discussion.
Page 15, line 48: You repeat the statement from line 9-14.	Thank you for your comment. Unfortunately we have been able to locate the statement
Discussion Page 17 line 23-30: I appreciate, that you want to illustrate the currently unutilized opportunity to address unmet vaccination needs in PEDs. But the uncertainty in your estimation of 5,000 potential vaccination moments in one year (period not mentioned) is not mentioned (see above).	Thank you for your comment. We have made it clearer that this is a maximum (estimated) figure.

Page 17, line 30-42: You explained, that vaccination rates are lowest in the age groups immediately after the recommended vaccination age and that there seems to be a catch-up effect with time. This may need to be considered for the comparison of your data and the COVER data.	Thank you for your comment. An additional sentence has been added to the results concerning comparison to clarify this point.
Page 18, line 5: I think that it may be interesting to discuss that the available methods to screen immunization status will vary depending on the country. Recall of immunization status is unreliable and handheld immunization records will most likely be unavailable in many PED consultations. In countries with Immunization registries or summary care records provide the opportunity to screen immunization status in a way that is not available in my country. I therefore need more information to understand what comprises inaccessible data? Were summary care records inaccessible?	Thank you for your comment. We have added an additional point to the strengths and weaknesses section to address this.
Since BMJopen addresses also international readers who might not be familiar with the (child) health care system in the UK, it might be helpful to provide more detail on the organization of vaccination campaigns (Are vaccinations mandatory? Are parents invited? Are there incentives? Do parents have to pay? etc.) and the documentation of vaccinations (Who does the documentation? Who keeps the health records? Who can access them? etc.).	Thank you for your comment. A paragraph detailing this information has been added to the introduction.
Is checking of vaccination status routinely part of treatment in PED?	Thank you for your comment. A paragraph detailing this information has been added to the introduction.
The benefits of using the PED for vaccinations enumerated in the introduction go very far. I suggest to focus more on the immediate benefits and on topics which are addressed by this study.	Thank you for your comment. We hope by addressing the comments throughout the review the focus is now clearer.
Page 8: "Within the scoping review...". Which scoping review?	Thank you for your comment. This sentence has been amended to clarify the previous study referred to.
The research question aims at demonstrating lower vaccination coverage in the PED compared with the general population. Please explain why you were interested in the difference of vaccination coverage.	Thank you for your comment. This paragraph has been amended to make this point clearer.
Please provide some information on the structure and data quality of health records.	Thank you for your comment. We have now added information about the summary care record to the introduction and methodology

	section. The second statement in strengths and weaknesses also discusses the issues with data quality and a separate paper detailing this further has been referenced (23).
Sample size calculation: What is the source for the PED prevalence of 0.77?	Thank you for your comment. Further information has now been provided.
Outcomes are described twice in the methods section. Please consider removing some of the information.	Thank you for your comment. Appropriate changes have been made.
Participants: "was roughly uniform." Please provide data to support this statement.	Thank you for your comment. A supplementary file has been referenced and provided to support this statement.
Estimate of "missing" tetanus and MMR vaccinations in a year: Please consider moving this paragraph to the discussion section.	Thank you for your comment, This paragraph has now been moved to the discussion.
Table 1: Pleas provide Ns.	Thank you for your comment. The table has been amended as requested.
Figure 1: bars for 3y3m to 3y6m: is the data correct? For most other age groups the black bars (children with tetanus and MMR vaccination) roughly correspond to the lowest bar within this age group (i.e. MMR vaccination).	Thank you for your comment. The data is correct. At age 3yr4months children become eligible for MMR2, yet only 2 had received this vaccine I this age band. We inserted an additional age bad (3y4m to 3y6m) to more accurately represent this as a third of the children in 3y3-3y6 were not actually eligible for MMR2 but were included in the denominator.
Figure 1: age band "total": are all categories (bars) applicable for the total sample?	Thank you for your comment. As discussed in the comment above the age band 3y4m-3y6m has not been included in the total figure as this would lead to double counting of these children.
The discussion covers many interesting and important topics. However, I miss the discussion of your main research question (difference in vaccination coverage between PED and general population).	Thank you for your comment. We have added a paragraph to the discussion detailing the discussion of the main research question.

The study's findings on great differences in tetanus and MMR vaccination prevalences is interesting. Do you have any ideas why MMR vaccination rate are lower?	Thank you for your comment. We do not know the reason for this finding and have added a suggestion for investigation in future studies.
The study period included the COVID-19 pandemic, including reduced access to health care in lockdown periods. Can you see potential lockdown effects in some age strata of your data?	Thank you for your comment, we have amended this sentence to add lockdowns as an example.